# Transcutaneous Electrical Acupoint Stimulation for Elders with Amnestic Mild Cognitive Impairment: A Randomized Controlled Pilot and Feasibility Trial

**DOI:** 10.3390/healthcare12191945

**Published:** 2024-09-28

**Authors:** Wenjing Xu, Zichun Ding, Heng Weng, Junyu Chen, Wenjing Tu, Yulei Song, Yamei Bai, Shuxia Yan, Guihua Xu

**Affiliations:** School of Nursing, Nanjing University of Chinese Medicine, Nanjing 210023, China; xwj_jy@163.com (W.X.); chrisdina2022@163.com (Z.D.); 837104@njucm.edu.cn (H.W.); 20223126@njucm.edu.cn (J.C.); 837017@njucm.edu.cn (W.T.); songyulei1986@126.com (Y.S.); czbym@njucm.edu.cn (Y.B.)

**Keywords:** mild cognitive impairment, transcutaneous electrical acupoint stimulation, cognitive function, randomized controlled trial

## Abstract

Background: Amnestic mild cognitive impairment (aMCI) is an important window of opportunity for early intervention and rehabilitation in dementia. The aim of this study was to investigate the feasibility and effect of delivering transcutaneous electrical acupuncture stimulation (TEAS) intervention to elders with aMCI. Methods: A total of 61 aMCI patients were randomly allocated into the intervention group (receiving a 12-week TEAS) and control group (receiving health education). The feasibility outcomes included recruitment rate, retention rate, adherence rate, and an exploration of patients’ views and suggestions on the research. The effective outcomes included cognitive function, sleep quality, and life quality, which were measured by the Montreal cognitive assessment scale (MoCA), auditory verbal learning test—Huashan version (AVLT-H), Pittsburgh sleep quality index (PSQI), and quality of life short-term-12 (QoL SF-12). Results: The recruitment rate, retention rate, and adherence rate were 67.35%, 92.42%, and 85.29%, respectively. Most aspects of the research design and administration of the TEAS intervention were acceptable. The quantitative analysis suggests that compared with the control group, the scores of MoCA, AVLT-H, and SF-12 (mental component summary) were significantly better (*p* < 0.05); however, the differences were not statistically significant in PSQI and SF-12 (physical component summary) (*p* > 0.05). Conclusions: The findings demonstrated that the study was feasible. TEAS awas possible for enhancing cognitive function and mental health in people with aMCI.

## 1. Introduction

Mild cognitive impairment (MCI) denotes a gradual deterioration in memory and cognitive abilities that does not impede daily functioning and falls short of the diagnostic criteria for dementia, which is the prodromal stage of Alzheimer’s disease (AD) [1]. MCI is further categorized into amnestic mild cognitive impairment (aMCI) or non-amnestic mild cognitive impairment (naMCI) based on the presence or absence of memory impairment. Studies suggest that aMCI may exhibit a greater propensity to progress to Alzheimer’s disease than naMCI [2]. Presently, aMCI is recognized as a crucial window of opportunity for early intervention and rehabilitation in dementia, offering an effective means to delay the progression of Alzheimer’s disease [3]. Consequently, identifying appropriate interventions to enhance cognitive function in individuals with aMCI remains a significant scientific challenge [4].

Nonetheless, consensus remains elusive regarding the efficacy of pharmaceutical interventions in enhancing cognitive function among individuals with aMCI [5]. Consequently, non-pharmacological therapies, such as Taichi, cognitive training, transcranial direct current stimulation, and so on, have gained prominence. Among these, electroacupuncture has demonstrated beneficial effects across a range of brain disorders, including anxiety amelioration [6], neurological rehabilitation [7,8], insomnia, and pain relief [8]. Notably, acupuncture has emerged as one of the most widely adopted adjunct therapies for MCI patients [9]. A meta-analysis supported the claim that acupuncture could ameliorate cognitive function in elderly individuals with MCI [10]. However, administering acupuncture necessitates skilled practitioners capable of precise and deep needling [11]. To overcome this, this study utilizes transcutaneous electrical acupoint stimulation (TEAS) as a non-invasive, painless, and superficial stimulation approach.

TEAS represents an innovative and practical technology that combines principles from meridian theory and transcutaneous electrical nerve stimulation (TENS), deeply rooted in traditional Chinese medicine [12]. While TEAS has shown promising results as an adjunctive treatment for various conditions, such as post-surgical cognitive impairment [13], there is limited research on its application in elderly individuals with mild cognitive impairment. Previous animal studies have indicated that TEAS exhibits similar effectiveness as electroacupuncture in slowing the progression of Alzheimer’s disease [14]. Therefore, the primary objective of this study was to determine the feasibility and acceptability of delivering TEAS intervention regimens three times a week for 12 weeks to elders with MCI. In addition, we aimed to assess the effects of the program on cognitive function (general cognitive function and memory, primary outcome), sleep quality (secondary outcome), and quality of life (secondary outcome) in TEAS compared to health education.

## 2. Materials and Methods

### 2.1. Study Design

To assess the feasibility of TEAS for aMCI, a pragmatic mixed methods randomized parallel-group exploratory design was implemented. This comprehensive approach involved evaluating the recruitment rate, retention rate, and adherence rate while also gathering insights and suggestions from patients undergoing the TEAS intervention. Simultaneously, interviews were conducted with patients who declined participation in the project to gain a deeper understanding of their reasons for reluctance. In addition to these feasibility considerations, the study also encompassed scientific assessments, including the impact of the TEAS intervention on patients’ cognitive function, sleep quality, and quality of life.

The study was conducted in accordance with the Declaration of Helsinki on the ethical principles for medical research involving human subjects. The study was approved by the ethics committee of Nanjing Hospital of Traditional Chinese Medicine, affiliated with Nanjing University of Chinese Medicine (KY2022004). The trial was registered in a Chinese trial register (trial-ID: ChiCTR2300075195).

Written informed consent was obtained from each participant.

### 2.2. Participants

The trial’s recruitment took place at Jingdongfang Care Centre in Jiangsu, China.

Petersen’s criteria [15] were referred to as the selection criteria. The inclusion criteria were as follows: (1) patients aged from 60 to 80 years old; (2) years of education ≥6 years; (3) scores of the mini-mental state examination (MMSE) ≥24; (4) scores of the Montreal cognitive assessment scale (MoCA) [16] <26 points (if the years of education ≤12 years, score plus 1 point); (5) scores of the clinical dementia rating scale (CDR) [17] of 0.5; (6) scores of the auditory verbal learning test-Huashan version (AVLT-H) [18] less than 1.5 standard deviations lower than the normal control; (7) Scores of Activities of Daily Living (ADL) [19] of ≤23; (8) normal visual hearing, able to conduct neuropsychological tests; (9) voluntary participation and signing of informed consent.

The exclusion criteria were as follows: (1) patients with a history of cerebrovascular disease, including ischemic cerebrovascular disease, hemorrhagic cerebrovascular disease, and neurological diseases such as hemiplegia and aphasia; (2) patients with secondary metabolic disorders caused by certain endocrine, genetic and nervous system diseases, and patients with metabolic diseases caused by drugs; (3) critically ill patients, patients with other central nervous system injury diseases or history, such as brain trauma, encephalitis, epilepsy, tumors, infections, severe liver and kidney dysfunction, blood system diseases, central nervous system demyelinating diseases, central nervous system degenerative diseases, etc.; (4) have received treatment to improve cognitive function in the past 3 months or are participating in other clinical trials; (5) patients with cardiac pacemakers.

### 2.3. Sample Size

As there were no previous clinical studies on the application of TEAS for intervening in patients with aMCI, the sample size for this study was estimated using an empirical method. Referring to previous studies [20,21], a sample size of 30 cases per group was considered adequate. To account for a 10% dropout rate, the final target was set at 33 participants per group, with a total sample size of 66.

### 2.4. Randomization and Blinding

Eligible participants were randomly assigned to the TEAS and control groups in a 1:1 ratio using a table of random numbers to generate the sequences. Allocation was concealed in envelopes opened by participants after completing the baseline assessment. Given the nature of the intervention and the single-site setting, it was not possible to avoid communication between participants or ensure complete blinding. However, outcome assessors, data collectors, and statisticians were blinded to the group assignments.

### 2.5. Intervention

TEAS was given 3 times a week for 30 min, and health education was given to each person in the intervention group. Only health education was given to participants in the control group.

#### 2.5.1. Health Education

Health education was provided to patients through lectures and the production of leaflets. The contents, which referred to practice guidelines [5] proposed by the American Academy of Neurology in 2018, included disease processes, risk factors, nonpharmacologic treatments, and so on. At the end of the experiment, supplementary treatment will be given to the members of the group.

#### 2.5.2. Transcutaneous Electrical Acupoint Stimulation

Transcutaneous electrical acupoint stimulation intervention for patients with aMCI was carried out by nurses who have systematically learned Chinese medicine nursing and TEAS operation. The acupoints selected in this study were Baihui (GV20), Shenting (GV24), Dazhui (GV14), and Shenshu (BL23) (alternated per time between left and right). The locations of these acupoints were determined based on the National Standard of the People’s Republic of China: Names and Positions of Acupoints for Localization (GB/T 12346–2021). The illustration of acupoint localizations is shown in Figure 1. The straps were equipped with Velcro for convenient fastening (see Figure 2A). In this particular setup, round sponge electrode pads measuring 5 cm in diameter (see Figure 2B) and saturated with saline were applied to specific points on the head. For the posterior regions, square hydrogel electrode sheets measuring 5 cm in width and 5 cm in length were utilized (see Figure 2C). These electrode pads were linked to an electronic and muscle instrument (SDZ-V, Hwato, Jiangsu Province, China), as illustrated in Figure 2D. The treatment protocol involved the application of sparse and dense waves with frequencies of 20/100 Hz, adjusting intensity according to the patient’s tolerance, and administering them for 30 min per session. The sessions occurred once every two days, three times a week, spanning a total duration of 12 weeks.

The procedural steps for TEAS were carefully executed as follows. Round sponge electrode pads were optimally moistened with saline, ensuring that the amount applied was sufficient without causing excessive dripping. The patient assumed a seated position, and the long strap was skillfully looped from the occipital ramus to the forehead, creating an encompassing circle. Two short straps were then utilized to securely affix the round sponge electrode pads to the designated acupuncture points. Next, Dazhui (GV14) and Shenshu (BL23) were systematically sterilized using 75% ethanol. The square hydrogel electrode slices were subsequently positioned on Dazhui (GV14) and Shenshu (BL23). The electrode slices were meticulously connected to the instrument using the corresponding leads. Finally, the unit was activated, with the selection of sparse and dense waves and the adjustment of intensity and duration to a 30 min session. Furthermore, prior to formalizing the procedure, each individual operator underwent an assessment. Only those who successfully passed the assessment were deemed eligible to participate as staff members in this study. Figure 3 shows the position of the electrodes.

### 2.6. Quantitative Study Outcomes

Before and at the end of the intervention, quantitative outcome indicators were measured.

#### 2.6.1. Feasibility Outcomes

The number of enrolled participants divided by the number of screened participants was the recruitment rate. The number of participants who completed the clinical trial divided by the number of enrolled participants was the retention rate. The number of participants who completed at least 28 TEAS treatments over the full course of treatment divided by the number of TEAS group participants was the adherence rate. Key feasibility parameters for the main trial were at least 70% recruitment, 90% completion, and 75% attendance at treatment sessions [22].

#### 2.6.2. Cognitive Function (Primary Outcomes)

MoCA served as the tool for evaluating global cognitive function, with a score range from 0 to 30. Scores are increased by 1 point (if <30) if participants have been out of education for more than 12 years. An overall score below 26 signaled cognitive decline and lower scores were indicative of more pronounced cognitive impairment [23]. The Chinese version of the MoCA demonstrated robust psychometric properties, with retest reliability of 0.857 and good internal consistency, as reflected by Cronbach’s α coefficient of 0.818.

AVLT-H, developed by Guo Qihao, was used to test the episodic memory function of older Chinese adults. The test began by assessing an individual’s ability to recall a list of 12 words over three learning trials, and the auditory verbal learning training (AVL-T) score was the sum of all correct responses given in the first three consecutive trials. Free recall was then administered after 5 and 20 min, respectively, which were short-term delayed recall (AVL-SR) and long-term delayed recall (AVL-LR). The higher the score, the better the delayed recall [18]. The internal consistency of the scale was 0.99, and the retest reliability was 0.87~0.94.

#### 2.6.3. Sleep Quality (Secondary Outcome)

The Pittsburgh Sleep Quality Index (PSQI) is composed of 19 self-rated questions and 5 questions rated by sleep peers. Only the 19 self-rated questions were scored; the 19 self-rated questions comprised a 7-factor scale ranging from 0 to 3. The cumulative score for each factor is the total score of the Pittsburgh Sleep Quality Index Scale, which ranges from 0 to 21, with higher scores indicating poorer sleep quality [24]. The Pittsburgh Sleep Quality Assessment is a commonly used tool for the subjective assessment of sleep quality in people with cognitive impairment [25,26]. A diagnostic sensitivity of 89.6% and specificity of 86.5% in the differentiation of good and poor sleepers in Chinese can be achieved with a global score of ≥6 [27]. The Cronbach’s α the Pittsburgh Sleep Quality Assessment in this study was 0.736.

#### 2.6.4. Life Quality (Secondary Outcome)

The short-term-12(SF-12) was used to assess the impact of health on an individual’s daily life. In this study, it was used to assess quality of life. The SF-12 included assessments of the role of vitality, social functioning, mood, mental health, general health, physical functioning, and physical and bodily pain. The first four indicators were used to assess the physical component summary (PCS), while the last four indicators were used to assess the mental component summary (MCS). Higher scores indicate better quality of life [28].

#### 2.6.5. Assessment of Safety

In the case of withdrawal before the end of the study, the reason for withdrawal was recorded. Patients were queried with open-ended questions regarding any adverse events experienced during each intervention to facilitate prompt action. Assessment and recording of the severity and adverse effects of the intervention would be carried out.

### 2.7. Data Analysis

#### 2.7.1. Quantitative Data

All statistical analyses were performed using IBM SPSS version 26.0. To ensure data integrity and accuracy, data entry was conducted independently by two researchers. For clinical trials, analyses were carried out according to the per-protocol analysis. Categorical variables in demographic information were expressed as n (%) and analyzed using chi-square tests. The Kolmogorov–Smirnov test was employed to assess the normality of continuous data. If data were normally distributed, they were presented as mean ± standard deviation (e.g., AVL-T, PSQI, PCS, and MCS). Otherwise, data were presented as median and interquartile ranges (e.g., MoCA, AVL-SR, AVL-LR). For normally distributed data (AVL-T, PSQI, PCS, and MCS), paired *t*-tests were used for within-group comparisons and two-sample *t*-tests for between-group comparisons. For non-normally distributed data (MoCA, AVL-SR, AVL-LR), the Mann–Whitney U test was used for between-group comparisons, and the Wilcoxon matched-pair signed rank test for within-group comparisons. Statistical significance was set at *p* < 0.05.

#### 2.7.2. Qualitative Data

The semi-structured interviews were conducted by the first author, who has had a systematic and in-depth study of qualitative research knowledge and interview skills. After obtaining informed consent from the participants, we conducted the interviews after the twelfth week of the intervention. The interview took place in a small, quiet room. Two preliminary interviews were conducted before the main study, but they were not taken into account during the analysis. The sample size is determined by information saturation to ensure completeness of data. No participants were asked to undertake a second interview, and no transcripts were returned to respondents for review.

The interviewer knew the participants’ group during the trial. The audio recordings were transcribed within 24 h of each interview. Thematic analysis was used to analyze the data inductively [29]. Researchers immersed themselves in transcribed texts, first identifying a series of codes, then comparing the codes and grouping them into subcategories, which in turn are abstracted into generic categories. Finally, the general categories were summarized into main categories. Two researchers independently analyzed the transcribed texts, and differences were resolved through discussion or consultation with a third researcher until full agreement was reached.

## 3. Results

### 3.1. Recruitment Processes and Resources

A screening of elderly individuals with amnestic mild cognitive impairment took place at the Jingdongfang Care Center on 1 September 2022. Patient enrollment started on 1 November 2022, marking the initiation of the recruitment process for seniors diagnosed with aMCI. A total of 334 individuals aged 60 years or older underwent assessment, with 98 meeting the criteria for mild cognitive impairment. Subsequently, 32 patients were excluded from the study for the following reasons: (1) not meeting inclusion criteria (n = 3); (2) declined to participate (n = 29). Thus, the recruitment rate was 67.35%. Following this, 66 patients were randomly assigned to either the TEAS group (n = 34) or the health education group (n = 32) (see Figure 2).

At the end of the quantitative study, the qualitative study was initiated on 1 September 2023. A total of 25 people participated in the qualitative interviews, and their characteristics are detailed in Appendix A. Among them, 13 patients diagnosed with aMCI underwent TEAS treatment. The interviews revealed that the primary motivations for elders’ participation were centered around improving their health, availing themselves of complimentary traditional Chinese medicine services, and contributing to research. Conversely, reasons for non-participation included challenges in allocating spare time, a limited understanding of the disease, skepticism towards TCM intervention, and a lack of trust in the research team.

“*My memory is getting worse and worse, especially worried that I will become senile dementia*”.[ID1]

“*You called me, and I have free time. And it was a free program, so I signed up*”.[ID13]

“*As a doctor myself, I prefer to support scientific research... I said yes as soon as I heard it was research*”.[ID11]

“*There is not much time available as there are usually multiple activities to attend. Therefore, participation is not always possible*”.[ID14]

“*I feel like I don’t have dementia. So, I don’t want to participate in it*”.[ID20]

“*I don’t believe in Chinese medicine and can’t accept this kind of treatment*”.[ID16]

“*I asked the children for their opinion. They are afraid that you are a salesman, so I do not participate*”.[ID19]

### 3.2. Intervention Management and Procedures

No issues relating to the acceptability of TEAS were identified. None of the participants had tried TEAS before. However, some participants believed that it belonged to traditional Chinese medicine treatment and had expectations for the results of treatment. Some participants were not sure if TEAS had a beneficial effect, and some said that TEAS had no effect at all.

“*I prefer Chinese medicine, which is always good for the body anyway*”.[ID7]

“*You said it was a trial. I’ll take it if it works*”.[ID5]

“*I don’t think it will work, but I’m willing to take part in this experiment to support you*”.[ID1]

Throughout the 12-week intervention period, two participants from the intervention group withdrew—one due to the installation of a pacemaker midway through the study and the other due to a change in residence. Simultaneously, three participants withdrew from the control group, citing the cumbersome nature of the testing process and their reluctance to repeat the tests (refer to Figure 4). The final retention rate stood at an impressive 92.42%. The adherence rate in the intervention group was commendable, with 29 patients completing TEAS treatments more than 28 times over the 3-month period, resulting in an adherence rate of 85.29%. Importantly, no adverse events were reported throughout the entire process.

Most participants reported that the TEAS treatment was comfortable without any adverse effects. Only one patient reported mild dizziness after treatment, which resolved on its own.

“*It feels good. I fall asleep every time I do it*”.[ID8]

“*Sometimes I feel dizzy after doing it, but I just need to rest for an hour or more*”.[ID4]

When asked why they could stick around to finish the trial, some said it was the good service from the researcher that kept them going; some said it was good for their health; some said they felt that their health was improving; others said that they had promised something that needed to be kept.

“*It must be that I want my body to be better, I can insist as long as it is good for the body. Also, you have such a good service attitude, many people can’t ask for that*”.[ID2]

“*I promised you I would finish it. If I have a promise to someone, I can do it*”.[ID5]

“*My lower back pain is much better after doing this... So, I want to keep doing it*”.[ID12]

When participants were asked for their opinions on the trial, some suggested increasing the duration of each intervention while reducing the number of sessions. However, the majority found the current schedule acceptable. There were also suggestions regarding health education, with some participants proposing the use of video and audio materials to make the instructions easier to understand and follow. Notably, a small number of older participants expressed interest in purchasing the equipment to perform the interventions at home, indicating a potential demand for home-based TEAS treatments.

### 3.3. Outcome Measures

Ultimately, 32 participants in the TEAS group and 29 participants in the control group were included in the analysis. There were no significant differences between the groups in terms of demographic or baseline characteristics (see Table 1).

In response to inquiries about their experiences during the measurements, a considerable number of participants mentioned feeling fatigued due to the extended duration of the tests. Some expressed a sense of unhappiness and mild distress upon realizing they couldn’t recall many words. On the other hand, a subset of individuals found the tests enjoyable. Some participants perceived the tests as tedious but conveyed a willingness to cooperate with the research team.

In evaluating treatment effects, participants’ responses were categorized into four aspects: cognitive function, psychological well-being, and physical health. Table 2 illustrates that the majority of participants acknowledged positive intervention effects, while three individuals perceived no significant change. Additionally, Table 2 provides a summary of two main categories, three general classes, and seven subclasses.

There was no significant difference between the TEAS group and the control group in overall MoCA scores before the intervention (*p* = 0.61). However, the overall MoCA scores differed significantly between the TEAS group and the control group after the intervention (*p* < 0.01). Total MoCA scores increased significantly after the intervention compared to before in the TEAS group (*p* < 0.01). However, in the control group, compared with those before intervention, there was no significant difference in the overall scores of MoCA after intervention (*p* = 0.37). Compared to the control group, there was also a significant difference in the difference score before and after the intervention in the TEAS group (*p* < 0.01) (see Table 3 and Figure 5).

In terms of AVLT scores, the difference between the two groups was not statistically significant before treatment (*p* > 0.05). After the intervention, the difference in the scores for AVL-T, AVL-SR, and AVL-LR between the TEAS group and the control group was statistically significant (*p* < 0.05). After treatment, patients in the TEAS group improved their AVL-T, AVL-SR, and AVL-LR scores compared to pre-treatment, and the difference was statistically significant (*p* < 0.05). The control group showed an increase in AVL-LR scores after the intervention compared to the scores before the intervention, and the difference was statistically significant (*p* = 0.01). However, post-treatment AVL-T and AVL-SR scores for patients in the control group did not change significantly compared with pre-treatment scores, and the difference was not statistically significant (*p* > 0.05) (see Table 4, Table 5 and Table 6 and Figure 5).

Regarding the PSQI scores, the difference between the two groups was not statistically significant before treatment (*p* > 0.05). However, PSQI scores were not significantly different between groups after intervention (*p* > 0.05). In the TEAS group, compared with before intervention, PSQI scores were significantly decreased after intervention (*p* = 0.01). In the control group, compared with before intervention, PSQI scores were not significantly different after intervention (*p* = 0.29) (see Table 7 and Figure 5).

In terms of SF-12 scores, the difference between the two groups before the treatment was not statistically significant (*p* > 0.05). After the intervention, the difference in the scores for MCS between the TEAS group and the control group was statistically significant (*p* < 0.01), but the difference in the scores for PCS between the TEAS group and the control group was not statistically significant (*p* > 0.05). In the TEAS group, compared with before intervention, PCS and MCS scores were significantly increased after intervention (*p* < 0.05). However, in the control group, compared with before intervention, PCS and MCS scores were significantly decreased after intervention (*p* < 0.05) (see Table 8 and Table 9 and Figure 5).

## 4. Discussion

To our knowledge, this is the first pilot feasibility study on the TEAS intervention in a Chinese population with amnestic mild cognitive impairment. The study achieved commendable recruitment, retention, and adherence rates at 67.35%, 92.42%, and 85.29%, respectively, aligning with the established criteria for crucial feasibility parameters [23]. This outcome attests to the viability of our approach. The majority of participants expressed satisfaction with the intervention sessions and the time required for treatment completion, underscoring the high acceptability of the treatment. Furthermore, our preliminary findings suggest a potential for cognitive function and mental state improvement in individuals with MCI. These results imply that the TEAS protocol could serve as a valuable complement to rehabilitation therapy for aMCI patients.

### 4.1. Reasons for Inclusion Criteria

The Mayo Clinic MCI criteria originally required people who did not have dementia to report memory complaints supported by objective deficits using episodic memory tests. Subsequent diagnostic criteria for MCI have all been expanded upon [30], with many drawing inspiration from Petersen’s 2004 proposal [15]. The inclusion criteria for this study were aligned with these diagnostic standards. In the context of dementia detection, MMSE demonstrated superior performance [31]. The optimal MMSE cut-off points for dementia screening were identified as 23/24 for individuals with seven or more years of education [32]. Consequently, MMSE ≥ 24 was adopted as a criterion to exclude patients with dementia. For the identification of MCI, the MoCA exhibited greater superiority compared to the MMSE [33]. Therefore, a MoCA score < 26 was considered indicative of Mild Cognitive Impairment in this study. Concurrently, the AVLT-H was employed to ascertain the presence of objective episodic memory impairment.

### 4.2. Reasons for Choosing TEAS

Chinese medicine has always had the advantage of “treating the disease before it is diagnosed,” and its non-pharmacological methods are characterized by “simplicity, convenience, inexpensiveness, and effectiveness”. Presently, various non-pharmacological therapies in Chinese medicine, including acupuncture, moxibustion, warm acupuncture, and acupressure, among others, are employed for the treatment of MCI. The non-pharmacological methods of Chinese medicine, particularly valued for their safety in comparison to pharmaceutical treatments, are progressively gaining prominence in clinical trials for addressing dementia and cognitive impairment associated with mild cognitive impairment (MCI) [34]. While acupuncture has a significant impact, its application requires a skilled professional. In contrast, TEAS is considered safer and more user-friendly. Additionally, acupressure, although self-administrable at home, demands a high level of self-motivation from the elderly [35]. In our study, the TEAS was operated by a nurse, but the elderly could also operate it themselves during the actual process. Mastery of locating acupoints and operating the device are the primary skills required. In future iterations, there is potential to simplify the equipment, particularly the head patch, which could be designed as a hat worn by the elderly. A simple switch mechanism could initiate the TEAS treatment, enhancing user-friendliness.

### 4.3. Reasons for Choosing Primary Outcome Indicators

Current tests for evaluating cognitive changes encompass various categories, including neuropsychological tests, biochemical measures, imaging tests, and electrical measures. The clinical ratings approach, widely acknowledged as a well-established and reliable method, is often deemed a ‘gold standard’ for the assessment and diagnosis of neurocognitive impairment in neuropsychological research and practice [36]. Two frequently utilized scales for detecting changes in overall cognitive function are the MoCA and the MMSE. This study preferred MoCA over MMSE due to its smaller ceiling effect, enhanced ability to detect cognitive heterogeneity, and a tendency toward improved diagnostic tracking [33,37]. As a result, MoCA was selected to assess changes in overall cognitive function in aMCI patients before and after the intervention. The AVLT is widely utilized in clinical neuropsychological testing for memory disorders due to its brevity, ease of administration, and straightforward scoring [38]. As a well-accepted tool in aging research, the AVLT is used to assess various memory indicators and processes [39]. In this study, AVLT was utilized to evaluate episodic memory function in older adults with MCI both pre- and post-intervention.

### 4.4. Feasibility of the Recruitment

Several factors contributed to the successful recruitment rate. Firstly, given the estimated prevalence of MCI at 15.5% [40], it became apparent that a large pool of individuals needed to be screened to identify sufficient MCI patients. To expedite this process, we established a partnership with the Jingdongfang Care Centre, a residence for hundreds of elderly individuals. This proved significantly more efficient than our previous approach of screening multiple communities in Jiangsu Province. Secondly, previous research indicated that awareness of MCI was limited, with only 18% of residents cognizant of it, 43% having never heard of MCI, and 55% equating it with ‘normal aging’ [41]. Consequently, it was challenging to encourage older individuals to accept the concept of MCI. To address this, we conducted a comprehensive pre-recruitment lecture to provide a systematic understanding of MCI, bridging this educational gap. Additionally, we noted that the care center regularly organized health education lectures, enhancing the cognitive foundation of its elderly residents regarding MCI. Lastly, the support of the care center engendered a high level of trust among the elderly population in our team, facilitating a smooth recruitment process.

### 4.5. Feasibility of the Intervention Management and Procedures

It is noteworthy that the study maintained a high retention rate and adherence rate, which may be attributed to several factors. Firstly, we ensured that participants were fully informed about the research protocol during initial contact, allowing them to make an informed decision regarding their participation. Secondly, our provision of door-to-door service for the elderly enhanced convenience and accessibility, facilitating their engagement in the study. Additionally, many elderly individuals at this care center did not reside with their offspring. In addition to administering the TEAS intervention, we also provided social companionship, which likely bolstered their motivation to stay involved. Moreover, a significant number of senior citizens participating in the study exhibited a high level of overall comprehension. Their cultural and health literacy levels were notable, enabling them to willingly support and cooperate with the research team’s arrangements.

In this study, TEAS was well accepted by participants. Initially, TEAS was met with limited resistance among most participants, in part due to the high confidence that Chinese seniors typically have in traditional Chinese medicine [42]. Despite some initial skepticism and negativity, participants experienced positive changes in their bodies during the intervention, leading to a subsequent shift in their attitude toward TEAS. Moreover, the majority of participants found the intervention enjoyable, with some considering purchasing their own machines for future self-interventions. While one participant reported dizziness after the intervention, this issue spontaneously resolved, confirming the safety of TEAS. These promising results suggest the potential for home-based TEAS interventions for patients with MCI in the future.

The reasons for participant dropouts were not linked to the TEAS intervention itself. In the control group, withdrawals were primarily linked to the outcome assessment process in this study. Some elderly participants expressed concerns about the test duration and experienced emotional distress when they could not recall certain words. For future studies, it might be advantageous to adjust the test duration, possibly dividing it into stages to alleviate participant fatigue. Additionally, proactive communication before the test can ensure that the elderly are aware of the test’s potential challenges in advance, fostering better participant understanding and engagement.

### 4.6. Benefits of TEAS on Cognitive Function

Elderly people with MCI had more memory problems than healthy elderly people. Therefore, the development of Alzheimer’s disease may be delayed or prevented by improving cognitive function, especially memory function, in people with aMCI. Therefore, alongside the assessment of global cognitive function, an additional test focused on memory function was incorporated. The outcomes revealed that the TEAS group demonstrated improvements, with higher scores in MoCA, AVL-T, AVL-SR, and AVL-LR compared to the control group. It showed that global cognitive function and episodic memory (including immediate recall and delayed recall) improved in aMCI patients in the TEAS group. The results were in concordance with previous findings of acupuncture in improving cognitive function in MCI or AD [43,44].

The mechanisms by which TEAS may improve cognitive function likely involve several neurobiological processes. One proposed mechanism is the activation of the Keap1/Nrf2 antioxidant stress pathway, which reduces oxidative stress damage, prevents neuronal injury, and inhibits neuronal apoptosis, thereby enhancing cognitive function [14]. Additionally, electroacupuncture has been shown to mitigate memory damage in the dorsal raphe nucleus of the brainstem, the initial site of intracellular neurofibrillary tangle lesions. This occurs through the reduction in tau hyperphosphorylation and the downregulation of glycogen synthase kinase-3β protein and mRNA expression levels [45]. From the perspective of traditional Chinese medicine (TCM), the pathogenesis of aMCI in the elderly is often attributed to kidney deficiency, essence deficiency, and marrow deficiency. The “Yishen Tiaodu” method, based on this pathogenesis, posits that benefiting the kidneys enriches the essence, which in turn nourishes the marrow, invigorates the yang qi, and facilitates the flow of spinal cord energy to the brain. This process is believed to improve cognitive function and mental clarity [46].

### 4.7. Benefits of TEAS on Sleep Quality

Sleep disturbance is a recognized risk factor for MCI5, and chronic sleep deprivation is a direct cause of memory loss [47]. Poor sleep quality interferes with neural plasticity in the brain, hindering the return of synaptic strength in memory-related brain cells to baseline levels [48]. Although the quantitative analysis in our study indicated a positive trend with improved PSQI scores in the TEAS group, the difference was not statistically significant when compared to the control group. However, qualitative data suggested that some participants experienced enhanced sleep quality.

Previous studies on acupuncture for insomnia have yielded mixed results. For example, Yin et al. [49] found that electroacupuncture significantly improved insomnia symptoms in depressed patients. However, other studies [50,51] suggested that acupuncture may be less effective in treating depression-related insomnia. These inconsistent results may stem from study design limitations, such as short treatment durations, varying acupuncture protocols, and differences in practitioner skills [52]. These factors could also explain the lack of significant improvement in sleep quality in our study. Future research should focus on optimizing acupuncture protocols for sleep quality enhancement and establishing standardized practices. Additionally, our study included older adults with and without pre-existing sleep disorders, introducing potential confounding factors that might have influenced the quantitative outcomes.

### 4.8. Benefits of TEAS on Life Quality

TEAS has shown potential benefits for health-related quality of life in elderly individuals with mild cognitive impairment. Cognitive decline often disrupts activity patterns, reduces social engagement, and leads to decreased physical strength and poor self-concept [53,54]. There was evidence from qualitative studies that patients felt more energetic and confident after the intervention. Qualitative feedback from this study revealed that participants felt more energetic and confident following the intervention. These findings are consistent with previous studies demonstrating that electroacupuncture can significantly alleviate symptoms of depression and anxiety [6,49,55,56]. The underlying mechanisms are thought to involve alterations in prefrontal cortical activity, changes in plasma levels of corticosteroids and adrenocorticotropic hormone, and modulation of platelet 5-HT levels [57]. These effects are likely mediated by the activation of the hypothalamic–pituitary–adrenal axis and the autonomic nervous system [58]. This may also explain the observed improvements in mental status following TEAS treatment.

Electroacupuncture has been shown to improve the quality of life in various conditions, particularly those related to pain and constipation [59,60]. However, the relationship between cognitive decline and quality of life is complex. Some studies suggest that cognitive decline does not necessarily impact the quality of life [61], while others indicate a strong association between the two [62]. The effect of cognitive decline on quality of life varies depending on the cognitive domain affected. Specifically, physical aspects of life quality tend to be less affected by cognitive impairment, whereas psychological aspects are more susceptible [63]. This aligns with our findings, where TEAS did not significantly improve the physical component of life quality.

Surprisingly, two participants reported a reduction in low back and waist pain. The results of this study were consistent with other studies that have found that transcutaneous electrical nerve stimulation can significantly reduce chronic low back pain and neck pain, as the selected acupuncture point, GV14, is close to the lower back, and BL23 is close to the waist [64,65].

### 4.9. Safety of TEAS

Mild dizziness was reported after treatment in one patient. The incidence of dizziness in patients receiving electroacupuncture has been reported as 4.3% in a previous study [55]. Another prospective observational study found that dizziness and fatigue occurred in 2.2% and 6.7% of those treated with dense cranial electroacupuncture stimulation plus body acupuncture, respectively [66]. No serious adverse events were reported, and the mild dizziness observed in our study subsided with rest. The occurrence of side effects should continue to be monitored in future trials, and if participants experience issues such as dizziness, the treatment should be paused to allow the patient to rest.

### 4.10. Limitations

This study has several limitations: (1) The absence of a sham stimulation control group may have introduced bias. The research was conducted in a nursing home where residents are mobile and can easily communicate with each other, making it challenging to administer a sham stimulation without compromising the blind. Consequently, we opted to use a health education intervention for the control group, consistent with similar studies [67,68]. To address this limitation in future research, trials could be conducted in different nursing homes to better ensure the integrity of a double-blind design. (2) Gender is a recognized risk factor for aMCI [69,70]. Although there were no significant differences in the demographic characteristics between the two groups in this study, future research should be designed more rigorously to investigate the effect of gender on the effectiveness of TEAS. (3) Three participants withdrew from the study due to the cumbersome nature of the testing process. In future studies, it may be beneficial to include adequate rest periods during testing and provide clear explanations to participants beforehand to maintain their enthusiasm and understanding. (4) Clinical neuropsychological testing quantifies the extent of cognitive deficits following neurological disorders [71]. Tan et al. [72] found that AVLT long-term delayed recall scores were associated with right frontal white matter N-acetylaspartate + N-acetylaspartylglutamate, which was negatively correlated. Meanwhile, MoCA and AVLT-H are classic tests for measuring cognitive and memory functions with high reliability and validity. Given these established associations and the reliability of neuropsychological testing, we believe it is reasonable to employ such assessments to make informed judgments about changes in cognitive functioning in older adults with amnestic Mild Cognitive Impairment (aMCI). However, the acknowledgment of the importance of enhancing the credibility of our findings in future studies was established. To achieve this, it is possible to incorporate objective indicators such as functional magnetic resonance imaging (fMRI), electroencephalography (EEG), and other relevant measures. (5) Feedback from participants suggests that future studies could benefit from diversifying the methods used for health education. Incorporating video and audio materials may enhance comprehension and adherence among elderly participants. Additionally, the interest expressed by some participants in performing the interventions at home highlights the potential for developing home-based TEAS protocols, which could be explored in future research.

## 5. Conclusions

The study findings indicate the feasibility of the research protocol and the acceptability of TEAS among participants. Moreover, the results suggest that TEAS holds promise for enhancing cognitive performance and quality of life in individuals with aMCI. These positive initial findings affirm the safety and effectiveness of TEAS, opening the possibility for its application in home care settings for those with AD or aMCI.

## Figures and Tables

**Figure 1 healthcare-12-01945-f001:**
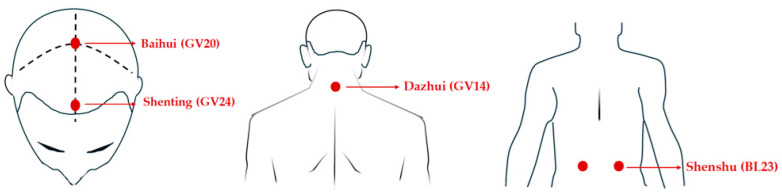
The location of the acupoints.

**Figure 2 healthcare-12-01945-f002:**
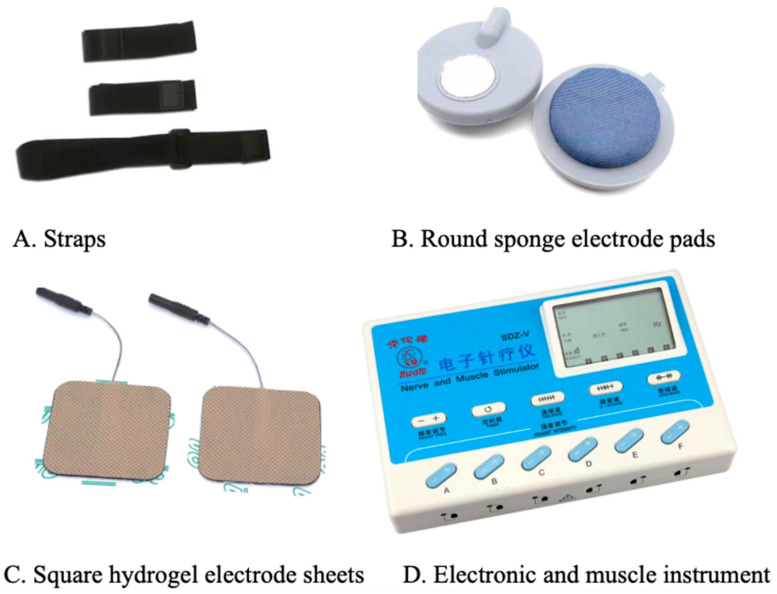
Instruments and accessories for transcutaneous electrical stimulation of acupuncture points.

**Figure 3 healthcare-12-01945-f003:**
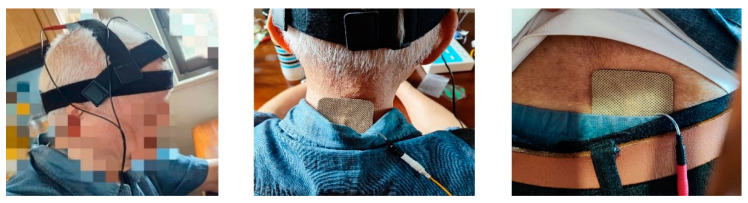
Position of the electrodes.

**Figure 4 healthcare-12-01945-f004:**
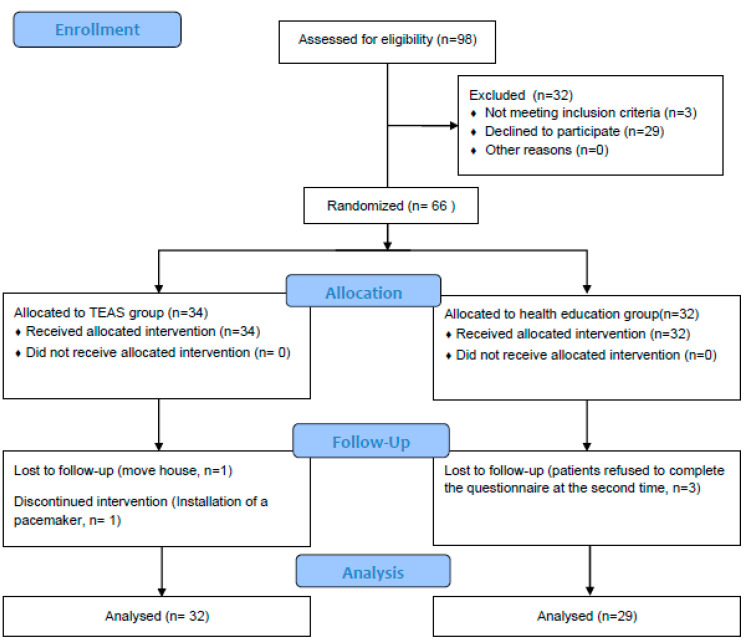
Study flow diagram.

**Figure 5 healthcare-12-01945-f005:**
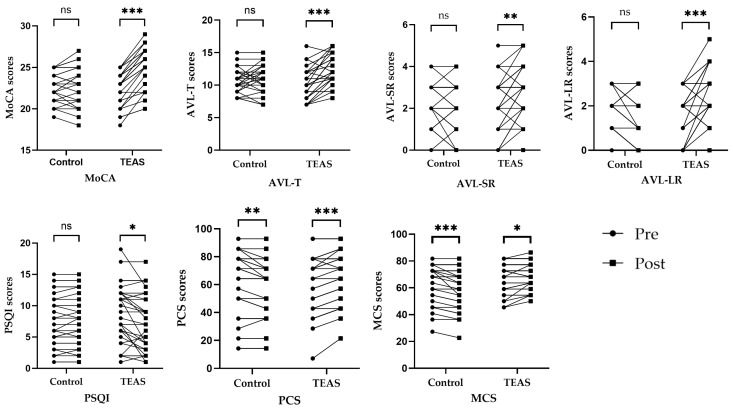
Dot plot of changes in outcome indicator scores in each group before and after the intervention. Notes. ns—*p* > 0.05 compared with the group before treatment. *—*p* < 0.05 compared with the group before treatment. **—*p* ≤ 0.01 compared with the group before treatment. ***—*p* ≤ 0.001 compared with the group before treatment.

**Table 1 healthcare-12-01945-t001:** Comparison of characteristics between the two groups.

	Control Group (n = 29)	TEAS Group (n =32)	*χ*^2^ (*Z*) Value	*df* Value	*p* Value
Age, Median (IRQ)	77(73, 79)	77(71.75, 78)	−0.015	/	0.988 *
Sex, n (%)			0.004	1	0.952 #
Male	10(34.5%)	13(40.6%)			
Female	19(65.5%)	19(59.4%)			
Years of education, median (IRQ)	13 (9.5, 16)	12(12, 16)	−0.844	/	0.398 *
Marital status, n (%)			0.272	1	0.602 #
Unmarried	0	0			
Married	22(75.9%)	28(87.5%)			
Divorced	0	0			
Widowhood	7(24.1%)	4(12.5%)			
Occupational nature, n (%)			0.029	2	0.986 #
Mental labor	25(86.2%)	27(84.4%)			
Manual labor	2(6.9%)	3(9.4%)			
Mental and physical labor combined	2(6.9%)	2(6.3%)			
Residence status, n (%)			1.298	1	0.255 #
Living alone	9(31.0%)	5(15.6%)			
Not living alone	17(58.6%)	27(84.4%)			
Living with spouse and children	1(3.4%)	0			
Living with friends	2(6.9%)	0			
Family history of dementia, n (%)			3.413	1	0.120 #
Denial	22(75.9%)	29(90.6%)			
Affirmation	7(24.1%)	3(9.4%)			

Note. * The *p* value was obtained using the Mann–Whitney U test. # The *p* value was obtained using the chi-square test.

**Table 2 healthcare-12-01945-t002:** Benefits perceived by participants.

Main Category	General Class	Subclass	ID	Representative Participant Quotes
TEAS effect	Cognitive function	Memory	7	“ I couldn’t name many people before, but I can remember their names now. It feels amazing”.
			13	“I used to go to the club and always forget to bring my thermos back, and now it’s less and less frequent”.
		Learning ability	8	“I feel like my brain is getting better. It used to take me a long time to understand your or others’ questions. Now I’m going to answer a little faster. That’s what my wife says”.
	Physical health	Sleep	6	“I used to have trouble falling asleep, but now I fall asleep quickly”.
			8	“It seems like I wake up less at night than I used to”.
			11	“I have a good night’s sleep now”.
		Pain relief	5	“I have cervical spondylosis. I used to have problems moving my neck... My neck feels much better now”.
			12	“I had terrible back pain. After doing this, my lower back pain has eased up a lot”.
		General state	2	“I have become more energetic than before, and I have some strength to do things”.
Extra care benefits	Mental state	Mood	6	“I was so happy when you came because you could talk to me”.

**Table 3 healthcare-12-01945-t003:** Comparison of MoCA scores before and after treatment.

	Pre	Post	*Z* Value	*p*1 Value
Control group	23.00 (21.50, 25.00)	23.00 (21.00, 25.00)	−0.892	0.373
TEAS group	23.00 (21.25, 24.00)	26.00 (24.25, 27.00)	−4.918	<0.001 *
*Z* value	−0.507	−3.568		
*p*2 value	0.612	<0.001 *		

Note. MoCA—Montreal cognitive assessment scale. Data were expressed in median (interquartile range). *p*1 value (by using Wilcoxon match-pair signed rank test) represents comparison before and after intervention within TEAS and control groups. *p*2 value (using Mann–Whitney U test) represents comparison between TEAS and control groups before/after intervention. *—*p* < 0.05.

**Table 4 healthcare-12-01945-t004:** Comparison of AVL-T scores before and after treatment.

	Pre	Post	*t* Value	*df* Value	*p*1 Value
Control group	10.62 ± 1.78	10.97 ± 2.13	−1.138	28.000	0.265
TEAS	10.19 ± 2.29	12.41 ± 2.14	−6.503	31.000	<0.001 *
*t* value	−0.818	2.633			
*df* value	59.000	59.000			
*p*2 value	0.417	0.011*			

Note. AVL-T—auditory verbal learning training test. Data were expressed in mean ± SD. *p*1 value (by using paired *t*-test) represents a comparison before and after intervention within TEAS and control groups. *p*2 value (by using two-sample *t*-tests) represents comparison between TEAS and control groups before/after intervention. *—*p* < 0.05.

**Table 5 healthcare-12-01945-t005:** Comparison of AVL-SR scores before and after treatment.

	Pre	Post	*Z* Value	*p*1 Value
Control group	2 (1, 3)	2 (1, 3)	−0.500	0.617
TEAS group	2 (1, 3)	3 (2, 4)	−3.305	0.001 *
*Z* value	−1.476	−1.086		
*p*2 value	0.140	0.278		

Note. AVL-SR—auditory verbal learning training test short-term delayed recall. Data were expressed in median (interquartile range). *p*1 value (by using Wilcoxon match-pair signed rank test) represents comparison before and after intervention within TEAS and control groups. *p*2 value (by using Mann–Whitney U test) represents comparison between TEAS and control groups before/after intervention. *—*p* < 0.05.

**Table 6 healthcare-12-01945-t006:** Comparison of AVL-LR scores before and after treatment.

	Pre	Post	*Z* Value	*p*1 Value
Control group	2 (0, 2)	1 (0, 2)	−0.504	0.614
TEAS group	2 (0, 2)	2 (2, 3.75)	−3.780	<0.001 *
*Z* value	−1.531	−1.981		
*p*2 value	0.126	0.048 *		

Note. AVL-LR—auditory verbal learning training test long-term delayed recall. Data were expressed in median (interquartile range). *p*1 value (by using Wilcoxon match-pair signed rank test) represents comparison before and after intervention within TEAS and control groups. *p*2 value (by using Mann–Whitney U test) represents comparison between TEAS and control groups before/after intervention. *—*p* < 0.05.

**Table 7 healthcare-12-01945-t007:** Comparison of PSQI scores before and after treatment.

	Pre	Post	*t* Value	*df* Value	*p*1 Value
Control group	6.9 ± 4.19	7.03 ± 4.25	−1.072	28.000	0.293
TEAS group	8.44 ± 4.72	7.31 ± 4.65	2.722	31.000	0.011 *
*t* value	1.341	0.243			
*df* value	59.000	59.000			
*p*2 value	0.185	0.809			

Note. PSQI—Pittsburgh Sleep Quality Index. Data were expressed in mean ± SD. *p*1 value (by using paired t-test) represents comparison before and after intervention within TEAS and control groups. *p*2 value (by using two-sample *t*-tests) represents comparison between TEAS and control groups before/after intervention. *—*p* < 0.05.

**Table 8 healthcare-12-01945-t008:** Comparison of PCS of SF-12 scores before and after treatment.

	Pre	Post	*t* Value	*df* Value	*p*1 Value
Control group	56.9 ± 23.5	54.19 ± 21.56	3.018	28.000	0.005 *
TEAS group	58.48 ± 21.19	63.62 ± 19.06	−6.411	31.000	<0.001 *
*t* value	0.277	1.813			
*df* value	59.000	59.000			
*p*2 value	0.783	0.075			

Note. PCS—physical component summary. SF-12—short-term-12. Data were expressed in mean ± SD. *p*1 value (by using paired *t*-test) represents comparison before and after intervention within TEAS and control groups. *p*2 value (by using two-sample t-tests) represents comparison between TEAS and control groups before/after intervention. *—*p* < 0.05.

**Table 9 healthcare-12-01945-t009:** Comparison of MCS of SF-12 scores before and after treatment.

	Pre	Post	*t* Value	*df* Value	*p*1 Value
Control group	62.7 ± 14.39	59.72 ± 15.16	3.931	28.000	0.001 *
TEAS group	67.61 ± 11.7	69.03 ± 10.35	−2.154	31.000	0.039 *
*t* value	1.470	2.825			
*df* value	59.000	59.000			
*p*2 value	0.147	0.006 *			

Note. MCS—mental component summary. SF-12—short-term-12. Data were expressed in mean ± SD. *p*1 value (by using paired *t*-test) represents comparison before and after intervention within TEAS and control groups. *p*2 value (by using two-sample *t*-tests) represents comparison between TEAS and control groups before/after intervention. *—*p* < 0.05.

## Data Availability

The data presented in this study are available upon request from the corresponding author. The data are not publicly available due to lack of permission from study subjects to access the data.

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
