# Peer review of "Transcutaneous Electrical Acupoint Stimulation for Elders with Amnestic Mild Cognitive Impairment: A Randomized Controlled Pilot and Feasibility Trial"

_healthcare, 2024, doi:10.3390/healthcare12191945_

Round 1
Reviewer 1 Report
Comments and Suggestions for Authors
The article under review is dedicated to use of electroacupuncture for cognitive impairment therapy.
The article is of middle length, well written easy to follow.
Introduction should be supplemented
Methods should be supplemented
Discussion shows study limitations
My points are:
1. My major concern is about study design. Can you advocate absence of “TEAS-placebo” group in which patients do not know did they get TEAS actually? Study of such therapeutic regimens should be double-blinded. Could you provide examples of similar studies?
2. In the introduction (maybe before “Notably, acupuncture…” in the 2nd paragraph) provide more refs on using TEAS for brain disease therapy. E.g. “Despite some reasonable skepticism (this may be omitted), circumstantial evidence shows the perspective of electroacupuncture for anxiety amelioration [10.1016/j.ctcp.2022.101541], can be used for neurological rehabilitation [10.1016/j.brainres.2023.148642, 10.1016/j.apmr.2017.03.023], and pain relief [10.1016/j.apmr.2017.03.023]. Notably, acupuncture…”.
3. In methods provide details of statistical analysis
4. Figure 1 or 2 – show the points you stimulated on the body scheme
5. In the results section please:
a) show the Table 3 data as figures with individual “progress” points (example figure is in provided pdf). This presentation will show score tendency for each patient and will strengthen the trust to your data.
b) show exact p values, t, and df in the table.
c) Table 3 legend “Wilcoxon signed rank test” – maybe “Wilcoxon match-pair signed rank test”? Please check.
6. Did you count effects on men and women separately? Are the differences in tests related to gender? Please check, maybe speculate on it in the discussion.
7. Appendix A – please check whether it is appropriate to publish the information like actual patient’s age and residence status. If I’m not mistaken, in NGS data deposition, they usually publish some “age range”. Maybe editor can help you.
Generally, I have concern about study design (comparison of effects in people who were “simply talked to” and in people who were “treated”) but suppose that the information obtained by authors is valuable and can be published after revision (statistics clarification and show the score data tendency for every patient).

Author Response
Dear reviewer:
We are very grateful for your constructive comments and suggestions for our manuscript entitled “Transcutaneous electrical acupoint stimulation for elders with amnestic mild cognitive impairment: A randomized controlled pilot and feasibility trial”. Your comments are very valuable and helpful for improving our manuscript. Our manuscript’s changes were all highlighted in yellow. In the following, the responses to all the comments are provided one by one.
We have tried our best to make all the revisions clear, and we hope that the revised manuscript can satisfy the requirements for publication.
Please see the attachment.

Reviewer 2 Report
Comments and Suggestions for Authors
The study is well-conducted and comprehensible in all its parts. Below, I offer a few minor suggestions for improving the paper.
It is noted that three participants withdrew from the control group, citing the cumbersome nature of the testing process and their reluctance to repeat the tests. I recommend adding this to the study's limitations and reflecting on alternative methods to prevent this in the future.
Additionally, I suggest a more thorough discussion of the potential side effects of the treatment. Even though the feasibility of the trial was demonstrated, possible side effects such as dizziness and fatigue should be addressed.
Author Response
Dear Reviewer:
We are very grateful for your constructive comments and suggestions for our manuscript entitled “Transcutaneous electrical acupoint stimulation for elders with amnestic mild cognitive impairment: A randomized controlled pilot and feasibility trial”. Your comments are very valuable and helpful for improving our manuscript. Our manuscript’s changes were all highlighted in yellow. In the following, the responses to all the comments are provided one by one.
We have tried our best to make all the revisions clear, and we hope that the revised manuscript can satisfy the requirements for publication.
Please see the attachment.

Reviewer 3 Report
Comments and Suggestions for Authors
This study presents a promising approach to cognitive enhancement in individuals with aMCI using TEAS. The use of a randomized controlled trial (RCT) enhances the validity of the findings, making it more likely that observed effects are due to the intervention rather than external factors. Moreover, the use of well-established and validated tools (MoCA, AVLT-H, PSQI, SF-12) for assessing cognitive function, sleep quality, and life quality strengthens the study's reliability and comparability with other research. Despite some limitations, the findings suggest that TEAS is a feasible intervention with potential cognitive and mental health benefits. With further research to address the following concerns, this line of investigation could contribute significantly to early dementia interventions.
1. The study includes a relatively small sample size (61 participants), which may limit the generalizability of the findings. Larger studies would be necessary to confirm the effects observed. Please clarify how the sample size is determined. Is power analysis applied?
2. Please clarify whether participants or assessors were blinded to the group allocation, which could introduce bias into the study outcomes. If blinding was not possible, discussing how this limitation was addressed would be beneficial.
3.The lack of significant differences in sleep quality (PSQI) and the physical component of life quality (SF-12) between the groups is acknowledged but not explored in detail. The manuscript would benefit from a discussion of potential reasons for these non-significant findings and how these findings might inform future research.
4.While the study demonstrates the feasibility and some effectiveness of TEAS, the mechanisms by which TEAS may improve cognitive function and mental health in aMCI are not addressed. Consider including a brief discussion on potential mechanisms by which TEAS may impact cognitive function and mental health in aMCI.
5. More information on the patients’ views and suggestions, and how these could influence the design of future studies, would add value to the manuscript.
Author Response

(The authors gave the same response as above.)

Round 2
Reviewer 1 Report
Comments and Suggestions for Authors
Authors responded all my comments properly.
I still have some concerns on study design but as authors showed the limitations of the study in discussion, I can recommend the study for publication.
I think that this study will provide new information on TEAS potential and suppose that detailed research is needed in the future.